

# Understanding pattern scaling errors across a range of emissions pathways

Christopher D. Wells[1,2], Lawrence S. Jackson[1,2], Amanda C. Maycock[1,2], Piers M. Forster[1,2]

[1]Institute of Climate and Atmospheric Science, University of Leeds, Leeds, UK
[2]Priestley International Centre for Climate, School of Earth and Environment, University of Leeds, Leeds, UK

*Correspondence to*: Christopher D. Wells (c.d.wells@leeds.ac.uk)

**Abstract.** The regional impacts of multiple possible future emission scenarios can be estimated by combining a few Earth System Model (ESM) simulations with a linear pattern scaling model such as MESMER which uses the pattern of local temperature responses per degree global warming. Here we use MESMER to emulate the future regional pattern of surface
temperature response based on historical single-forcer and future Shared Socioeconomic Pathway (SSP) CMIP6 simulations. Pattern scaling errors are decomposed into two components: differences in scaling patterns between scenarios, and intrinsic timeseries differences between local and global responses in the target scenario. The timeseries error is relatively small for high-emissions scenarios, contributing around 20% of the total error, but is similar in magnitude to the pattern error for lower-emission scenarios. This irreducible timeseries error limits the efficacy of pattern scaling for emulating strong mitigation
pathways and reduces the dependence on the predictor pattern used. The results help guide the choice of predictor scenarios and where to target introducing other dependent variables beyond global surface temperature into pattern scaling models.

## 1 Introduction

Anthropogenic climate change has already driven significant impacts throughout the globe, and these will continue to become
more severe (IPCC, 2021). Estimates of the impacts of future emissions depend on several sources of uncertainty – internal variability, model structural uncertainty, and unknowns in the emissions themselves (Hawkins & Sutton, 2009). The first two sources of uncertainty are often explored using multi-member ensembles of Earth System Models (ESMs). The emissions uncertainty can only be explored by constructing multiple potential future emissions scenarios, and investigating their respective impacts (Riahi et al., 2022).

The most recent generation of Integrated Assessment Model (IAM)-simulated scenarios are the Shared Socioeconomic Pathways (SSPs) (Gidden et al., 2019). These have been used in ESMs to assess future climate response in IPCC AR6 WG1 (Lee et al., 2021). ESMs are computationally expensive to run, so in general they can only simulate a handful of future emission scenarios (O'Neill et al., 2016, Tebaldi et al., 2021). The small number of scenarios used for ESM simulations can mask uncertainties in future pathways as the full range of plausible emissions parameter space is not adequately sampled (Grubler
et al., 2018; Otero et al., 2020; Partanen et al., 2018).



Both the IPCC WGI and WGIII AR6 reports used simplified physical climate model emulators trained on more complex ESM simulations to assess possible future projections of global surface temperature (Lee et al., 2021; Forster et al., 2021; Riahi et al., 2022; Kikstra et al., 2022). However, the assessment of regional climate projections relied largely on simulations from ESMs and Regional Climate Models (RCMs). Since the 2015 Paris Agreement, which frames international climate goals in

terms of global temperature targets, there has been growing emphasis of examining climate change impacts as a function of warming level (IPCC, 2018). In IPCC AR6, future scenarios simulated by ESMs were compared at different global mean temperature levels (Lee et al., 2021; Seneviratne et al., 2021). The suitability of this approach depends on the extent to which regional changes scale with global temperature change, which in turn depends on the variable of interest and the details of the GHG and aerosols scenario. Regional emulation tools operate on a similar basis, using techniques such as pattern scaling to

translate simple large-scale information such as global mean temperature into estimates of the spatially resolved climate response for a broad range of scenarios (James et al., 2017; J. F. B. Mitchell et al., 1999; Osborn et al., 2018).

These approaches can be limited by systematic response variations between and within scenarios. Long-lived species' forcings are independent of their emission location, but the impacts of short-lived species such as aerosols strongly depend on their emission region, with typically larger effects locally (L. Liu et al., 2018; Persad & Caldeira, 2018). The patterns of climate

change impacts also differ between transient and equilibrium climates – the transient temperature response is typically larger over land than over ocean, due to the greater thermal inertia of the latter (Herger et al., 2015; Huang et al., 2020; King et al., 2020; T. D. Mitchell, 2003). These two factors – variations in the forcer pattern and the level of disequilibrium – can be expected to break the linearity assumed within pattern scaling if they differ between and within scenarios (Goodwin et al., 2020; Herger et al., 2015; James et al., 2017; T. D. Mitchell, 2003; Osborn et al., 2018). Pattern scaling has been modified to

partially address these issues using patterns varying between forcers (Kravitz et al., 2017; Xu & Lin, 2017; Schlesinger et al., 2000) and response timescale (Zappa et al., 2020).

Other nonlinearities can occur, especially under higher emissions scenarios (Lopez et al., 2014), for example the removal of sea ice in the Arctic will saturate the ice-albedo feedback and reduce the local temperature sensitivity (Huang et al., 2020; Lynch et al., 2017), though higher sensitivity may initially occur due to sea ice thinning (Ishizaki et al., 2012). Some responses

of the climate system to external forcing such as sea-ice retreat and intertropical convergence zone shifts move geographically and will therefore be poorly represented by pattern scaling (Herger et al., 2015).

Despite the potential limitations of pattern scaling for climate emulation, the technique has been shown to work well across a range of scenarios (Alexeeff et al., 2018; Beusch et al., 2020; J. F. B. Mitchell et al., 1999). A linear approximation has been found to be reasonable for regional temperature changes within and between sets of scenarios (Seneviratne et al., 2016;

Seneviratne & Hauser, 2020), with similar response patterns found within different emissions scenarios in the near-term (Lee et al., 2021). The variation in response patterns between scenarios is typically less than the variation between models, indicating that the errors arising from pattern scaling are smaller than model uncertainty (Goodwin et al., 2020; Herger et al., 2015; Osborn et al., 2016, 2018; Tebaldi & Arblaster, 2014; Tebaldi & Knutti, 2018). Pattern scaling errors are generally substantially larger when "extrapolating" (projecting a scenario with higher forcing than the data used to generate the pattern) than when



interpolating (Herger et al., 2015; T. D. Mitchell, 2003; Tebaldi et al., 2020), with smaller errors for more modest forcing differences between the predictor and target scenarios (Osborn et al., 2018).

Pattern scaling has been used to emulate regional changes in temperature (Beusch et al., 2020; Link et al., 2019), to forecast temperature and precipitation simultaneously (Snyder et al., 2019), and to study changes in extreme precipitation (Thackeray et al., 2022). The application of pattern scaling to precipitation is complicated relative to temperature by the larger role of

internal variability (Hawkins & Sutton, 2011), the presence of strong nonlinearities and local factors (G. Liu et al., 2022), and the role of forcing-specific adjustments (Myhre et al., 2018). However, extreme precipitation is more closely constrained by moisture availability and may be more successfully emulated through pattern scaling (Pendergrass et al., 2015; Sillmann et al., 2017). Pattern scaling has also been incorporated with Earth system components such as land surface models to make faster projections (Zelazowski et al., 2018). It has been applied to estimate the regional effects of single-country greenhouse gas

emissions in order to attribute their local temperature impacts (Beusch et al., 2022), and to estimate the country-level economic impacts attributable to each other country's $CO_2$ emissions (Callahan & Mankin, 2022). These exercises would require vast computer resources and time if they were attempted with ESMs.

The scenarios to which pattern scaling has generally been successfully applied typically have smaller inter-scenario variation in aerosol emissions and warming rates than more recent, and likely future, scenarios of interest. Many historical studies

applied pattern scaling to the CMIP5-era RCPs (Alexeeff et al., 2018; Goodwin et al., 2020; Herger et al., 2015; Ishizaki et al., 2012; Kravitz et al., 2017; Lynch et al., 2017; Osborn et al., 2018; Tebaldi et al., 2020; Tebaldi & Arblaster, 2014; Xu & Lin, 2017), which all exhibit similar decreases in anthropogenic aerosol emissions into the future (Gidden et al., 2019), a much narrower range than projected among the newer SSP scenarios used in CMIP6. This may make the SSP scenarios less amenable to pattern scaling than prior scenarios (Goodwin et al., 2020). The SSP scenarios include SSP119, approaching the 1.5°C level

under the Paris Agreement (Meinshausen et al., 2020), with stronger mitigation than the RCPs. Many low-emissions Paris agreement-consistent scenarios were assessed as part of IPCC AR6 WGIII (IPCC, 2022), but relatively few have been systematically studied in multi-ESM projects. Low-emission scenarios may also encounter issues due to sampling variability in the pattern-generation regression, and stabilisation scenarios may be more susceptible to physical nonlinearities (Osborn et al., 2018). Indeed, many studies that find pattern scaling to be accurate with earlier scenarios note that under stronger mitigation

or wider ranges in aerosol emissions, the technique would be less effective (Alexeeff et al., 2018; J. F. B. Mitchell et al., 1999; Tebaldi & Arblaster, 2014).

While tools such as MESMER (Modular Earth System Model Emulator with spatially Resolved output) have been developed to implement pattern scaling using the SSP scenarios (Beusch et al., 2020), a systematic analysis of the range of errors associated with the application of pattern scaling to temperature within the SSPs remains to be done. Multi-model studies

analysing pattern scaling efficacy for low-emissions scenarios are also lacking (Tebaldi & Knutti, 2018). The effect of the choice of predictor data used to generate the pattern utilised in pattern scaling has also not been fully explored. This paper takes steps to address these gaps, through a novel decomposition of pattern scaling errors into their relation to pattern scaling assumptions.




The reliance on linearity under pattern scaling can be split into two key assumptions, relating to space and time:

The Pattern Assumption: Pattern Scaling assumes that the pattern of change is constant between all scenarios, regardless of the mix and level of forcers within them.

The Timeseries Assumption: Pattern Scaling assumes that the timeseries response at each location follows the same shape as the global response, simply modified by the local sensitivity (i.e. the pattern), thus allowing the local change timeseries to be estimated by scaling the global timeseries by a constant local pattern coefficient. This can be thought of as assuming the pattern

is constant in time within a given scenario.

These two assumptions will have associated errors when the method is applied - this decomposition is described further in Section 2.2. This paper studies the effects of the pattern scaling assumptions on regional temperature projections, and decomposes the pattern scaling error into components due to each of these, exploring how these errors vary when projecting different emissions scenarios. The effect of different choices of (one or more) predictor scenario(s) is also tested to determine

the impacts of this decision on emulation accuracy. Inter-model variation is investigated for all the impacts studied.

Section 2 sets out the ESM data utilised and the model used to perform the pattern scaling analysis, and shows how to diagnose the error associated with each pattern scaling assumption. Section 3 presents the key results, Section 4 explores the implications for the application of pattern scaling, and Section 5 provides discussion and conclusions.

## 115 2 Methods

### 2.1 Data and pattern scaling approach

Two sets of emissions pathways are the focus of this paper. To understand the effect of different forcers on the warming pattern and pattern scaling errors, two historical (1850-2020) scenarios from the Detection and attribution MIP (DAMIP) (Gillett et al., 2016) project are used: hist-aer, which includes only anthropogenic aerosol emissions, and hist-GHG, which includes only

greenhouse gas emissions. This allows for an idealised comparison of the difference of the pattern attributable to historical levels of different forcers, although neither represents a realistic emissions pathway due to the co-emission of aerosols and GHGs in reality. To determine the difference between warming patterns amongst coherent emissions pathways, the SSP scenarios are used, examining several SSPs but focussing on SSP119 and SSP585.

Data for all scenarios were taken for annual mean temperatures from the cmip6-ng database (Brunner et al., 2020), which re-

grids all data to a common 2.5°x2.5° latitude-longitude grid to allow for inter-model comparison. For each of the two sets of emissions pathways, all models with at least one member of each experiment were used: 10 ESMs for the two DAMIP scenarios and eight for the five SSPs. The models and number of members for each scenario are given in Supplementary Tables S1 and S2.

This study utilises the mean response component of the MESMER model (Beusch et al., 2020), implementing pattern scaling

to emulate the spatial annual mean temperature response in a scenario. Pattern scaling generates more accurate emulations





than the timeshift method, which selects a window of data centred around the year in which the global average reaches a desired Global Warming Level (Tebaldi et al., 2020). The pattern is derived from linear regression of the local response on the global response at each gridcell. This linear regression approach to pattern scaling has been shown to provide more accurate patterns than the alternative "delta" method, whereby the average climate towards the end of a scenario is subtracted from that in the early period (Lynch et al., 2017; T. D. Mitchell, 2003). In the default configuration of MESMER, the raw annual gridcell-level data is regressed against the smoothed global temperatures, but here this is modified to use the same smoothing on both local and global temperatures. This is performed to ensure the global average parameter is very close to 1 K/K, as it should be by definition, when predictor the model on an individual low-emission scenario such as SSP119. The smoothing performed is Locally Weighted Scatterplot Smoothing (LOWESS), which takes the weighted average of the timeseries across a moving window. The weighting is tricube, and the window fraction set to the default MESMER value of 50 divided by the number of timesteps as in the default case – when using the SSP data here, this is 50/85 = 0.6. MESMER's default version only includes land gridpoints, to focus on land impacts, but here all gridpoints were used, to study the broader response. The intercept of the linear regression, zero in theory and generally small in magnitude in practice (Beusch et al., 2020), is added to the emulation in MESMER.

A given emulation consists of the predictor set – comprising one or many scenarios – and a target scenario. The pattern is derived via the linear regression of the local on global temperature anomalies, relative to the first 50 years of each predictor scenario. This pattern is then multiplied by the global temperature timeseries of the target scenario to generate the emulation. The difference between this emulation and the actual ESM pattern is defined as the pattern scaling error. The pattern scaling error is zero in the global mean by design – since the pattern (with average value 1) is simply scaled by the global temperature in the ESM – but errors occur regionally, and the global average of the local absolute error will therefore not be zero.

## 2.2 Decomposing pattern scaling errors

As described in Section 1, the pattern scaling error can be thought of as deviations from two key assumptions: the pattern and timeseries assumptions. Short-term inter-annual variability is dampened via the LOWESS smoothing, though decadal-scale variability will also be present.

Figure 1 shows a schematic demonstrating the decomposition of pattern scaling errors into the pattern and timeseries errors. An idealised scenario is shown whereby temperatures relative to pre-industrial times rise from 1 K in 2015 to 2 K, and fall back to 1 K by 2100. The four different timeseries used to apply and evaluate pattern scaling are shown - the local and global responses in both the target and predictor datasets. The "local" response applies to an arbitrary location. Pattern scaling determines the local parameter from the regression of the local on global predictor data, and scales the target global temperature by this value. The pattern scaling error is then the difference between this projection and the actual local response in the target scenario:

Pattern error = Total error - Timeseries error.



The top row of Figure 1 shows "perfect emulation", whereby all four of these timeseries are identical. The scaling parameter is therefore equal between the scenarios (left plot), and the shape of the emulated response is identical to the target dataset (middle), with zero error at all times (right).

The second row shows the effect of altering the shape of the local timeseries, but keeping the warming parameter (i.e. the pattern) the same. The scenarios are still identical - the global response is the same for both, as is the local response - but the

shape of the local and global responses within each scenario now differs. In this case, since the pattern is still correct, the time-mean error is zero, but not at each timestep due to this breakdown in the linearity between local and global scales. This error can therefore be understood as the "timeseries error"; that is, the error due to the differing local and global timeseries response within the scenario.

The third row now alters the scenarios in a different way. The local and global responses are now identical within each

timeseries, but the scenarios themselves are different - the local parameter of the predictor is larger than in the target. Since the local and global timeseries in the target scenario follow the same shape, the error is purely due to differences in the scaling parameter, i.e. the local value of the pattern. This error is therefore the "pattern error".

Finally, the fourth row applies both of these changes simultaneously; the local parameter differs between the scenarios, and the local and global timeseries differ within each scenario. This is the general case, which can be expected in real-world and

ESM data due to nonlinearities within the climate system. The total error is comprised of both pattern and timeseries errors, but the split between them is not clear from the timeseries. The two errors happen to roughly cancel out in the early decades, but have the same sign in later years.

The general total error, from using one scenario to emulate another, can be decomposed into the pattern and timeseries errors. The timeseries error, in row two of Figure 1, was generated by using the same scenario as the predictor and the target, termed

"self-emulation". The error is due to the internal dynamics of the response; specifically, the difference between the shape of the local and global temperature timeseries. This error is therefore intrinsic to the target scenario. For a given predictor-target pair, the timeseries error can be found by calculating the target-target pattern scaling error, i.e the error upon "self-emulation" of the target. This can then be subtracted from the original predictor-target pattern scaling error to determine the pattern error. The contribution of each error to the total error can then be studied; the error timeseries in the bottom row of Figure 1 shows

this decomposition applied to the idealised scenarios. The idealised scenarios in Figure 1 exhibit some symmetries in the timeseries and pattern errors; this is not necessarily the case in practice.





Figure 1: Demonstrating the pattern scaling error decomposition for several idealised scenarios (see text for details). Left: local temperature timeseries against global. Middle: timeseries of the target scenario local response and the emulated local projection. Right: pattern scaling error timeseries i.e. the difference between the emulation and the target scenario in the Middle figure.





## 3 Results

### 3.1 Effect of pattern error

The first pattern scaling assumption – that the pattern is independent of the predictor scenario – is investigated in this section, first in the DAMIP experiments and then in the SSP scenarios.

The top row of Figure 2 shows the multi-model mean hist-aer and hist-GHG response pattern based on regression across the whole period (1850-2020). Note that while aerosols drive a cooling, since the local response is regressed against the global mean the sign cancels, with the pattern giving the sensitivity under warming or cooling. The dominant canonical pattern of the hist-GHG response is greater sensitivity over land than ocean, expected due to the lower heat capacity of the land surface and lower capacity for evaporative cooling (Byrne & O'Gorman, 2018; Lee et al., 2021). The Arctic exhibits a strong amplification (Holland & Bitz, 2003). In hist-aer, the land-ocean distinction is still clear, but the northern hemisphere land is particularly sensitive, due to the historical concentration of aerosol emissions within this region. The bottom panels in Figure 2 show the multi-model mean difference between the patterns, and the magnitude of this difference minus the inter-model standard deviation. The difference between the patterns is over 0.5 K in the Northern hemisphere mid-latitudes (NHMLs), where hist-aer is more sensitive. Because the pattern averages to 1 globally, the more sensitive NHMLs in hist-aer lead to less sensitive areas more remotely, over the southern ocean and Antarctica. The strongest of these errors are larger than the typical errors between patterns found in prior work, which are usually around 0.4 K or less (Huang et al., 2020; Ishizaki et al., 2012; Lynch et al., 2017; J. F. B. Mitchell et al., 1999; T. D. Mitchell, 2003), but this is to be expected due to the complete separation of forcers – and their associated patterns – here. There is substantial inter-model variability, but broad areas still see a difference larger than one inter-model standard deviation (i.e. red regions). Parts of the NHMLs exhibit a significantly more sensitive response to hist-aer, including the USA, Europe, and east Asia, and the Southern Hemisphere oceans are significantly less sensitive. Despite the Indian subcontinent experiencing a large magnitude difference (more sensitive to aerosols), the large variability in the aerosol sensitivity (see Supplementary Figure S1), potentially due to model variation in the monsoon response to aerosols, leads to no inter-model agreement on this.




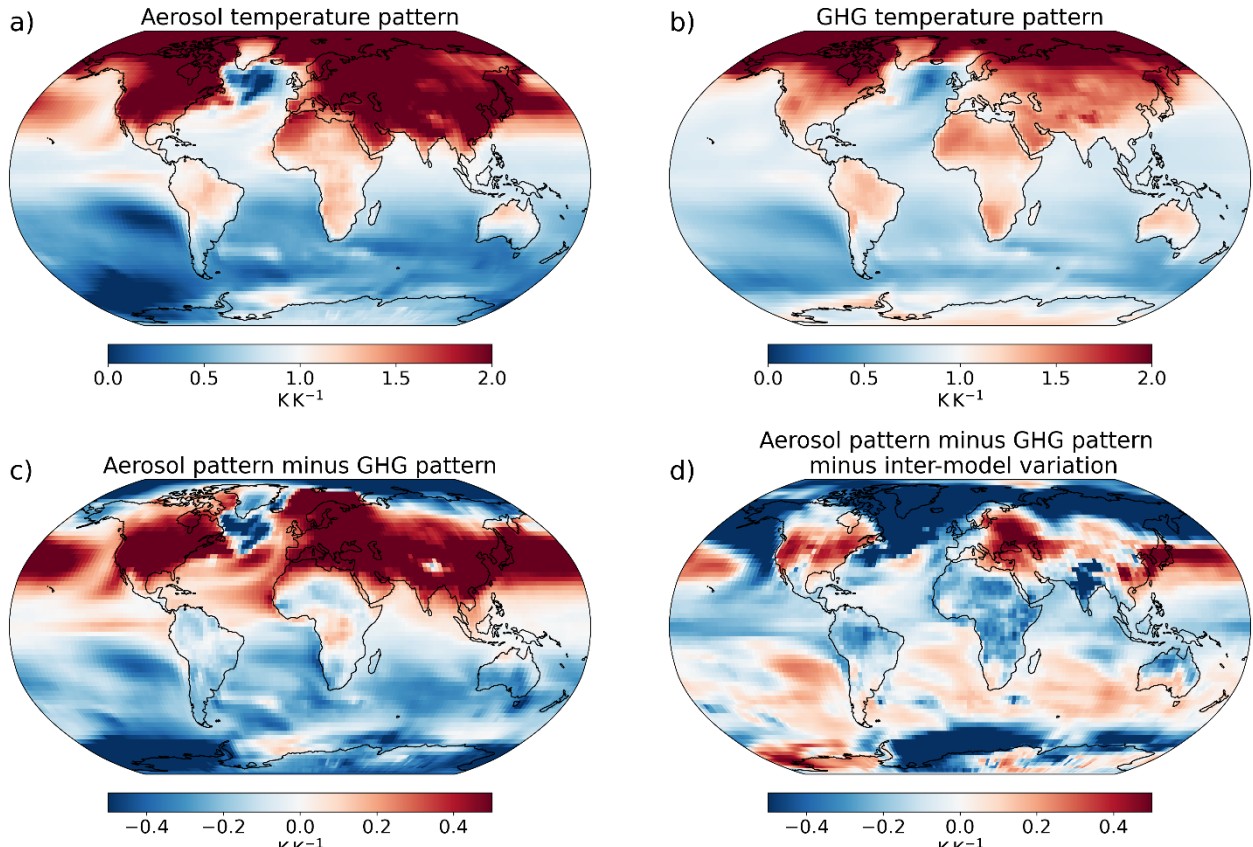

Figure 2: Mean warming patterns across 10 ESMs derived from historical GHG-only (hist-GHG; top left) and aerosol-only
(hist-aer; top right) simulations, the aerosol pattern minus that from GHGs (bottom left), and the magnitude of this difference
minus the inter-model standard deviation in this difference (bottom right).

Figure 3 shows the same analysis for SSP119 and SSP585 in a similar way to Figure 2. SSP585's pattern shows greater
sensitivity over land, consistent with the higher warming rate maintained through the century in this scenario, and a less
sensitive Arctic, likely due to a saturation of the Arctic sea ice feedback (Huang et al., 2020; Lynch et al., 2017). Overall the
pattern difference is similar to the transient minus equilibrium patterns found in previous studies (Herger et al., 2015; Huang
et al., 2020; King et al., 2020; T. D. Mitchell, 2003). This indicates that the difference between spatial patterns of warming in
SSP119 and SSP585 may be driven primarily by the differences in disequilibrium rather than aerosol emissions; aerosols can
be expected to play a relatively larger role in the pattern difference between scenarios closer in radiative forcing and/or with
larger aerosol emissions differences. The lower temperature sensitivity in East Asia under SSP585 may be linked to the weaker
reduction in aerosols there than in SSP119, resulting in less "unmasking" of the cooling effect. Some features of the SSP585-
SSP119 pattern difference vary from the RCP8.5-RCP2.6 differences found by Ishizaki et al. (2012), who found a more





sensitive Arctic under the higher emissions scenario. This they suggested may be attributable to stronger ice melt overall under RCP8.5, due to thinning of the sea ice under warming. This highlights the contingency of the local sensitivity on the baseline
climatology; further analysis of these background conditions in the ESMs may aid in explaining the differences (Lynch et al., 2017), but this is beyond the scope of this paper. Comparing to Figure 2, the pattern differences are typically smaller than those between hist-GHG and hist-aer, with most areas seeing differences less than 0.3 K, consistent with differences between plausible scenario patterns in prior studies (Huang et al., 2020; Ishizaki et al., 2012; Lynch et al., 2017; J. F. B. Mitchell et al., 1999; T. D. Mitchell, 2003).

Compared to the DAMIP comparison, the SSP119 and SSP585 pattern difference is not as robust between models, with fewer regions having differences greater than one inter-model standard deviation (Figure 3d compared to Figure 2d). This is expected due to the narrower differences in forcers between scenarios and consistent with prior work on pattern differences between plausible scenarios (Goodwin et al., 2020; Herger et al., 2015; Osborn et al., 2016, 2018; Tebaldi & Arblaster, 2014; Tebaldi & Knutti, 2018). Supplementary Figure S2 shows the analysis for all the combinations of the five SSPs analysed in this study;
generally, differences between SSPs closer in radiative forcing show fewer significant differences in their spatial patterns, in agreement with Osborn et al. (2018). Clear, significant differences are therefore found between the temperature response patterns attributable to different historical forcers, consistent with their different spatial patterns. These differences are less systematic across ESMs for the future scenarios analysed, which are likely relatively more affected by their differing warming rates.






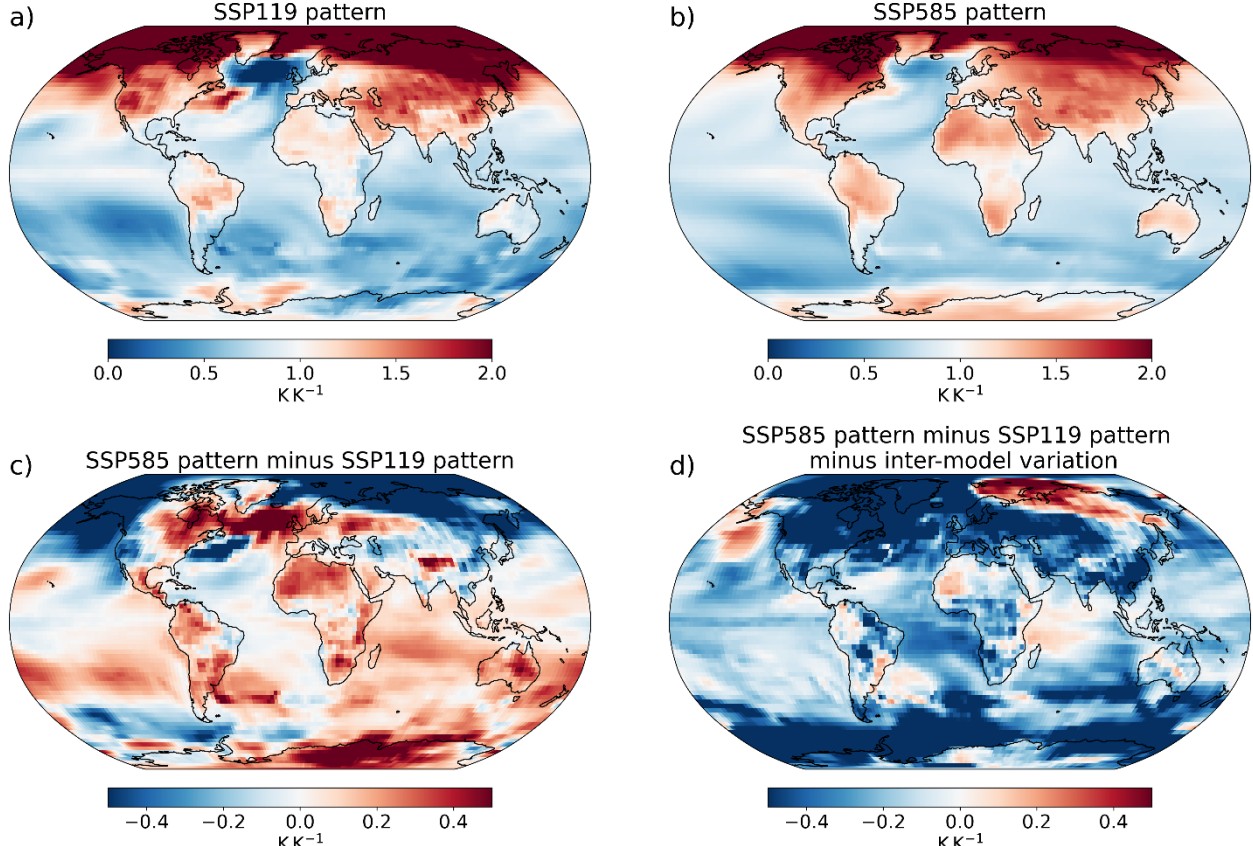

Figure 3: Mean warming patterns across 8 ESMs derived from SSP119 (top left) and SSP585 (top right) simulations, the SSP585 minus that from SSP119 (bottom left), and the magnitude of this difference minus the inter-model standard deviation in this difference (bottom right).


## 3.2 Total pattern scaling error

Overall errors in emulation when using the pattern from each of the two DAMIP experiments (hist-aer and hist-GHG) to separately emulate both experiments – with four combinations in total, two self-emulations and two out-of-sample – are displayed in Figure 4 for 1990-2020 in the multi-model mean. The last 30 years are shown to indicate the errors that arise once
a substantial forcing has been applied, and to study the self-emulation scaling error within the period, as self-emulation scaling errors cancel over the whole period.

Out-of-sample errors are generally substantially larger than self-emulation (Osborn et al., 2018), since out-of-sample emulations introduce pattern errors in addition to the timeseries error. The out-of-sample emulations are both too warm over the NHMLs and too cool in the southern hemisphere, in keeping with the pattern differences. In the hist-aer-hist-GHG





emulation (using the hist-aer pattern to emulate the hist-GHG response), the warming in the NHMLs is overestimated, since the aerosol pattern is more sensitive here than the GHG response, and the Southern Ocean is conversely under-sensitive. In the hist-GHG-hist-aer emulation, although the anomaly is positive over the NHMLs, this represents an underestimation of the cooling (since both the emulation and ESM data are negative). This is due to the lower sensitivity of the GHG response here.

Variation in the local and global timeseries shapes can be due to spatial variations in the forcing, or in the response. Self-
emulation hist-GHG errors, attributable to timeseries variation within the hist-GHG response, are small, indicating there is little internal timeseries variation in this experiment. This is consistent with the well-mixed nature of GHGs. There will still be physical non-linearities, in both the concentration-forcing and forcing-response mechanisms, and internal variability, which are reflected in the non-zero self-emulation scaling errors, but these are small in magnitude. The largest feature is an oversensitivity in the Arctic, which may be due to a saturation of the ice albedo feedback.

In the hist-aer self-emulation, by comparison, while still small compared to the out-of-sample errors, some coherent errors occur. Negative anomalies (overestimated cooling) occur in the NHMLs, with positive ones (underestimated cooling) over the tropics and south Asia. This indicates that the sensitivity of the NHML temperature to the global change is lower in 1990-2020 than the average across the period, since the local parameters calculated across the entire timeseries are too strong at this time. This is consistent with a shift in aerosol emissions from the NHMLs to Asia over the last 3 decades, and explains the positive
anomalies over Asia – the emulation is under-sensitive here in this period since aerosol emissions are more concentrated in Asia in 1990-2020 than on average through the period.  This is validated by Supplementary Figure S3 showing the reverse effect in the mid-20th century (as NHML aerosol emissions were historically concentrated locally in this period) and the earlier peaking of NHML temperatures than the global average in hist-aer.

Pattern scaling errors in a given period can therefore be related to the differences in the average pattern between the predictor
and target scenarios, and the internal dynamics of the target scenario, particularly due to spatial variations in the forcing. Note that the timeseries self-emulation scaling error is also included in the total out-of-sample error, though the relative size indicates the pattern error is the dominant factor – these relative sizes are explored in more detail in Section 3.4.

The pattern scaling errors in Figure 4 are typically less than 0.2 K for self-emulation, and over 0.5 K for out-of-sample projection in the NHMLs. This out-of-sample error is substantial compared to the simulated temperature change of around 1.5
K/ -0.5 K globally in hist-GHG/hist-aer, and 2 K/-1 K in the NHMLs. As for the pattern differences, these out-of-sample errors are larger than typically found under pattern scaling (Alexeeff et al., 2018; Herger et al., 2015; J. F. B. Mitchell et al., 1999; T. D. Mitchell, 2003; Osborn et al., 2018; Tebaldi & Knutti, 2018), especially considering the magnitude of global warming in the scenarios, due to the starker differences between the forcing patterns in these idealised experiments.



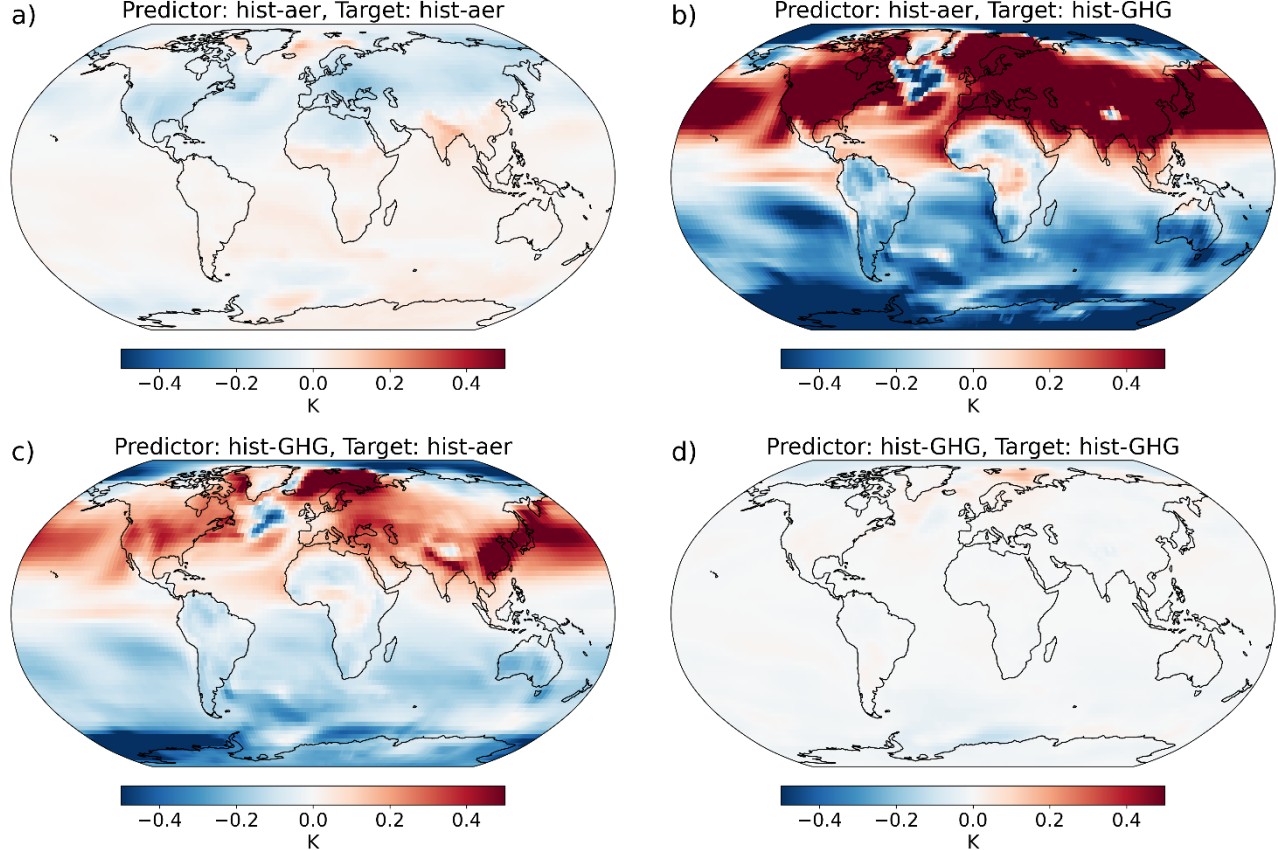

Figure 4: 1990-2020 pattern scaling errors (emulation minus ESM) averaged across 10 ESMs when predicting with historical GHGs and aerosols separately, and targeting the historical GHG and aerosol response separately; four combinations in total.

The magnitude of the errors in Figure 4 minus the inter-model standard deviation in the error are shown in Supplementary Figure S4. Errors are significant in the out-of-sample emulations, over both the NHMLs and the Southern Ocean, consistent with the significant pattern differences in Figure 2. The hist-GHG self-emulation shows no coherent errors, due to the lack of pattern error and coherent timeseries variation, but the NHML errors in the hist-aer self-emulation are consistent between the models, indicating agreement on the timeseries error found here.

Similarly, Figure 5 shows the 2070-2100 multi-model mean pattern scaling errors under the four combinations of the SSP119 and SSP585 scenarios (self-emulation and cross emulation), and Supplementary Figure S5 shows the magnitude of these minus one standard deviation of model spread. The out-of-sample errors are again larger than those attributable to self-emulation alone, indicating a substantial role of the pattern difference. They are consistent with the pattern differences in Figure 3 – more sensitive tropical land and less sensitive Arctic in SSP585 generating consequent errors in both out-of-sample cases – however, similarly to the pattern differences, the inter-model standard deviation is generally larger than the magnitude, with significant





errors generally only in those areas with significant difference in the patterns themselves. The timeseries (self-emulation) error is small in SSP585, similarly to hist-GHG, but SSP119 shows larger errors. Specifically, an error pattern consistent with the pattern differences occurs. This timeseries error, as with hist-aer, must be driven by the internal characteristics of SSP119. The period 2070-2100 exhibits less positive warming trends than the timeseries average, with peak warming occurring in mid-century on average. Thus, the parameters derived from the timeseries average will be too sensitive over land and tropical

oceans, as the climate is experiencing less positive forcing than average, similarly to the SSP585-SSP119 pattern difference. This leads here to broad, coherent self-emulation scaling errors over 2070-2100, but as for the out-of-sample cases there is little inter-model agreement. Consistent results are found for each pair of the five SSPs used here, shown in Supplementary Figures S6 and S7; extrapolation to project higher-forcing scenarios using lower-forcing patterns is found to introduce substantial errors, with interpolation to lower-forcing scenarios generating smaller errors, though still larger than self-

emulation due to the additional effect of pattern errors (Herger et al., 2015; T. D. Mitchell, 2003; Tebaldi et al., 2020). The errors under self-emulation are typically less than 0.3 K, with strong interpolation less than 0.5 K but strong extrapolation over 0.5 K in broad areas. This compares to around 1.4 K/4.4 K warming relative to pre-industrial times in SSP119/SSP585 over 2081-2100 (IPCC, 2021). These self-emulation and interpolation errors are consistent with those found in prior work (Alexeeff et al., 2018; Herger et al., 2015; J. F. B. Mitchell et al., 1999; T. D. Mitchell, 2003; Osborn et al., 2018; Tebaldi & Knutti,

2018), while the extrapolation errors are larger due to the extreme case study showcased in this section.



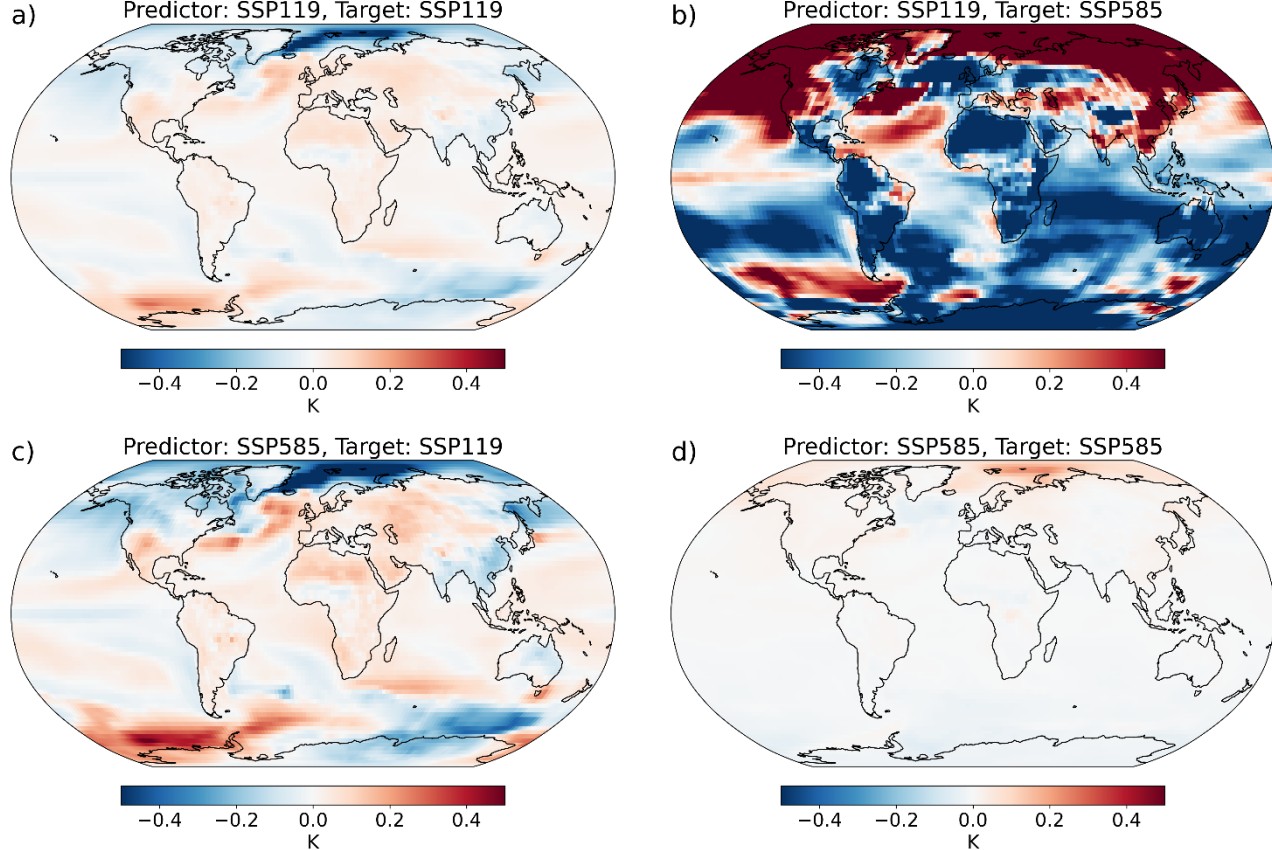

Figure 5: 2070-2100 pattern scaling errors (emulation minus ESM) averaged across eight ESMs when predicting with SSP119 and SSP585 separately, and targeting the SSP119 and SSP585 responses separately; four combinations in total.

## 3.3 Relative importance of the pattern and timeseries errors

As highlighted above, it is important to understand how the magnitude and relative size of the two types of pattern scaling error – pattern and timeseries – depend on the target and predictor dataset.

Pairwise predictor-target emulations are performed for the 25 combinations of the five SSPs analysed here. Maps of the timeseries and pattern errors are both calculated in each year of the simulation. Pattern scaling errors are zero on the global average by design – as the pattern is scaled by the global mean response – so to analyse the size of the errors, the global average of the error magnitude is taken for each. The magnitude of the total error is also taken – this is equal to the timeseries error for self-emulation, but for out-of-sample emulations, local cancellations from opposite sign timeseries and pattern errors cause this total error to be less than the sum of the two. This sum of the two – termed the sum error – is also calculated to allow for comparisons between the two magnitudes.





Figure 6 shows the 2015-2100 timeseries of the pattern and timeseries errors for the four combinations of SSP119 and SSP585.
Each line gives the multi-model mean, with shading indicating plus and minus one inter-model standard deviation. Note the
varying vertical scales. Supplementary Figure S8 gives the results for each of the 25 SSP combinations, with a fixed scale.
The timeseries error, dependent only on the target scenario, varies relatively little between scenarios, while the pattern error is
substantially larger for extrapolation cases. Upon even only adjacent extrapolation – e.g. using SSP126 to emulate SSP245 –
the pattern error becomes increasingly large to 2100.

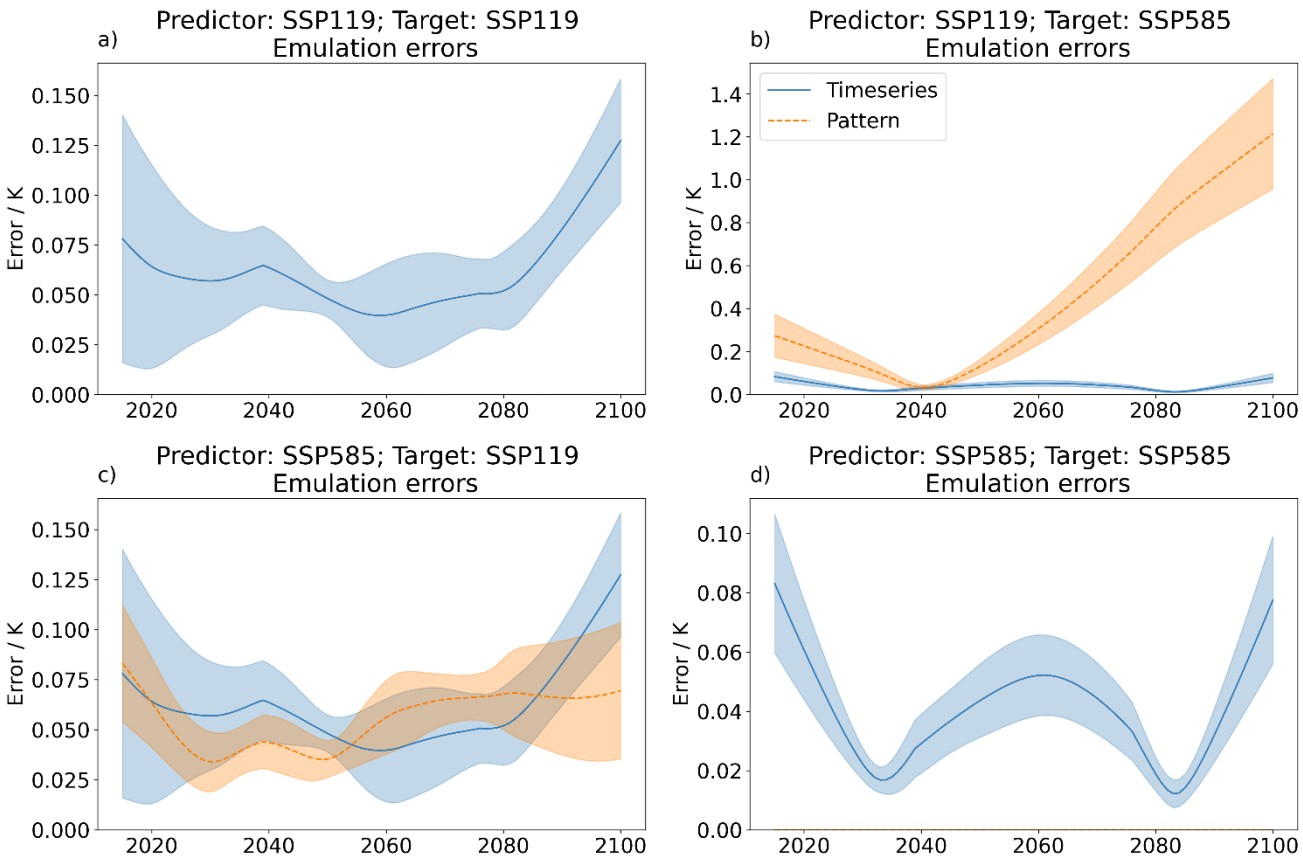

Figure 6: Timeseries of the size of the global average pattern scaling error attributed to pattern errors and timeseries errors.

The magnitude of the time-averaged pattern, timeseries, and total errors in each pair is shown in Figure 7 for the 25 scenario
combinations, along with the percentage of the sum error (the sum of the magnitudes of each component) given by the pattern
error. The sum of the error magnitudes is used to compare the magnitudes, as opposing sign errors will partially cancel in the
total. Note the smaller scale on the timeseries error plot. The magnitude of the timeseries error is less than 0.1 K in all scenarios,
but largest in SSP119, a scenario in which this error can represent a significant fraction of the mean response. The pattern





error, zero for self-emulation by definition, is systematically higher for extrapolation, likely strongly influenced by the scaling
of the pattern difference by the target scenario global mean timeseries. Global and time-averaged pattern error magnitudes can
reach almost 0.5 K under the highest extrapolation (SSP119-SSP585), but are still around 0.2 K for slighter extrapolations and
0.1 K under interpolation. The total error is therefore highest for extrapolation. There is less dependence of the total error on
the predictor scenario when targeting a low emissions scenario than targeting a high one, due to the greater role of the timeseries
error – note the slight variations in the SSP119 column compared to the SSP585 one.

Pattern errors represent a different amount of the sum errors under different pairs, accounting for over 80% under strong
extrapolation, but only around half for projecting SSP119. The intrinsic timeseries error, irreducible under this methodology,
accounts for a much larger fraction of the error under low-emissions scenarios than the pattern error. This larger role of
timeseries error is consistent with the lower correlation between local and global temperatures under low emissions scenarios
found by Lynch et al. (2017).





Figure 7: Pattern scaling errors averaged over the scenario time period for each predictor-target pair; errors are calculated annually, the absolute value taken, and then averaged across time and models. Top left shows the pattern error; top right the timeseries error (with a smaller scale); and bottom left the total error. Bottom right gives the percentage of the absolute total error (the sum of pattern and timeseries) attributed to the pattern error.

**3.4 Effect of peak warming on timeseries error**

The year of peak warming is increasingly used to classify emission scenarios (Riahi et al. 2022). One implication of simple pattern scaling approaches tied to global warming level is that if, in a low-emission scenario, the global warming timeseries





peaks in a particular year, then the emulated temperature peaks in this same year in every gridpoint. The spatial pattern of this peak warming is then homogeneous by design. Any spatial structure in the peak warming pattern of the actual ESM target data will be missed, leading to pattern scaling errors.

   The effect of this can be tested by exploring the peak warming simulated in ESM simulations of low-emission scenarios. Supplementary Figure S9 shows the multi-model mean of the deviation in the local year of peak warming from the global

average, along with the magnitude of this deviation minus the inter-model standard deviation, for both SSP119 and SSP126. Generally, tropical land and oceans peak earlier than average, and the Arctic and Southern Ocean later, consistent with the inertia of the system as seen in Figure 3. The patterns are similar between SSP119 and SSP126, indicating some consistency between scenarios in this effect. Few areas see inter-model agreement, however, with agreement on earlier peaking over some tropical oceans and land, and later peaking over the east of the Southern Ocean.

Figure 8 shows results from three region-model-scenario combinations to demonstrate the effect this can have on emulation errors. The global and regional ESM timeseries are shown in each case, along with the self-emulated local trend, all LOWESS-smoothed. Perfect emulation would occur if the orange and blue dashed lines - the local ESM data and the local emulation - overlapped; deviations from this are the pattern scaling error. The top left shows the Arctic - EC-Earth3 - SSP119 case. Consistent with the spatial map for this model in Supplementary Figure S10, in the ESM the global temperature peaks in mid-

century, but the Arctic continues to warm to 2100. Since the pattern scaling projection is simply scaled by the global mean, however, the Arctic emulation peaks with the global temperature, diverging from the ESM to the end of the century, and projecting the wrong sign of trend from mid-century onwards.

   The top right of Figure 8 applies this to the "North Atlantic Warming Hole" (NAWH) in MRI-ESM2-0 under SSP126 – the area of the North Atlantic where the general warming has been masked by circulation change-induced local cooling historically

(Huang et al., 2020). In the ESM, the NAWH cools throughout the century, while global temperatures peak around 2070. The pattern scaling parameter here is negative, as local cooling is regressed onto overall global warming. The projected NAWH response therefore reaches a minimum when the global mean peaks, and warms from there to 2100 as global temperatures reduce.

   Finally, the bottom left of Figure 8 applies this to Europe in MRI-ESM2-0 in SSP126. In the ESM data, European and global

temperatures rise, stabilise, and fall, but as a land region near the NAWH, Europe peaks in temperature several decades before the global mean. As shown in the bottom right of Figure 8, these different regional and global shapes happen to produce a linear fit with almost exactly zero gradient – although Europe sees substantial temperature change through the century, the average sensitivity averages to zero due to the differing global peak time. The resultant emulation thus has essentially zero amplitude, deviating significantly from the substantial changes modelled over Europe in the ESM.


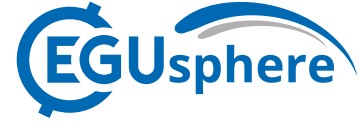

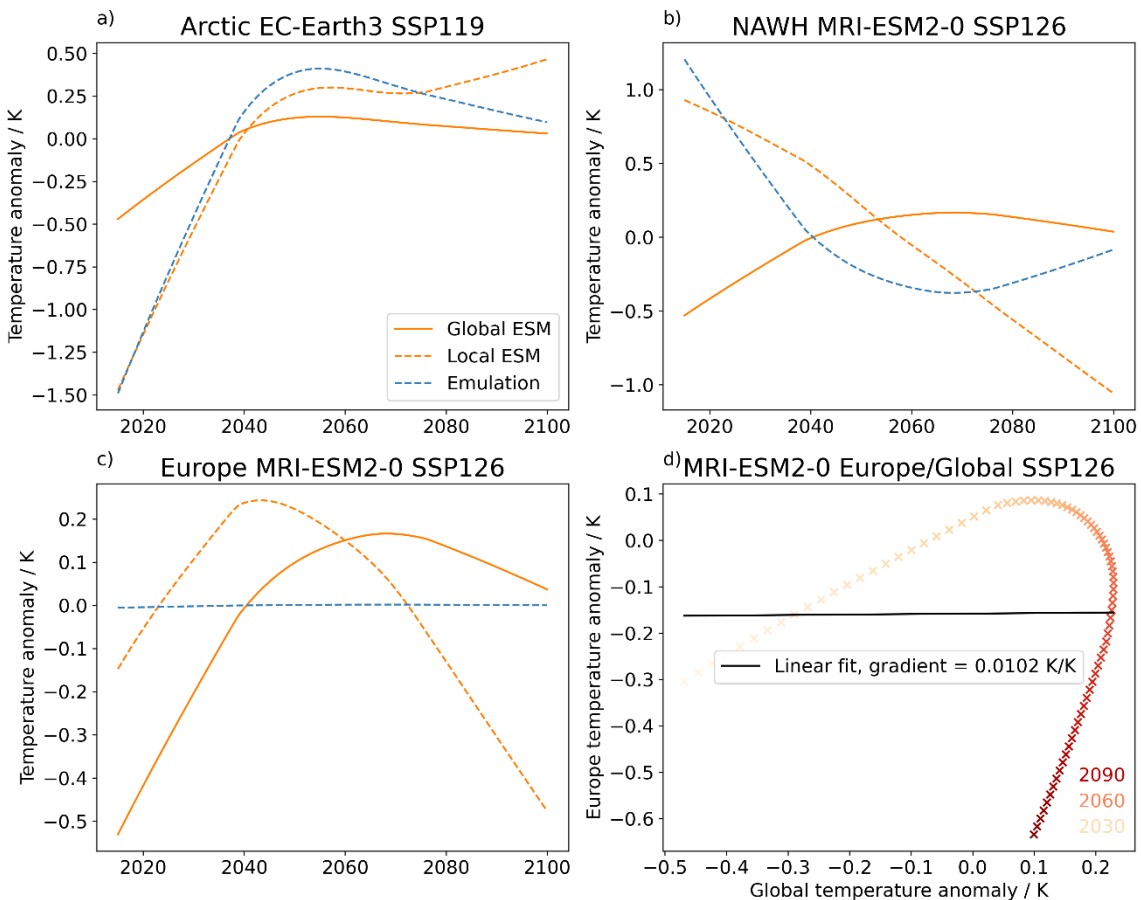

Figure 8: Timeseries of global and regional ESM annual temperature changes (relative to the first 50 years of a scenario), both annual and LOWESS smoothed, and the projected regional emulation from fitting a linear regression to the local and global smoothed temperatures, for three region-model-scenarios: Arctic (60° N to 90° N) in EC-Earth3 under SSP119, and the North Atlantic (40° N to 50° N, 40° W to 20° W) and Europe (35° N to 70° N, 10° W to 40° E) in MRI-ESM2-0 in SSP126. The local and global peak warming years are indicated with dashed and solid vertical black lines respectively. Also shown is the regression curve for the third example.

## 4 Implications for the use of pattern scaling

The implications of these findings for the application of the pattern scaling methodology are studied in this section.





When emulating a given scenario, a choice must be made as to the predictor dataset used – the scenario(s) used to generate the
       pattern. The effect of this choice on pattern scaling errors is important to understand; it won't affect the timeseries error by
       definition, but it will modify the pattern error and hence the overall emulation efficacy. Four options are studied here, for
       targeting both SSP119 and SSP585 separately; some include the target scenario in the predictor dataset, thus not being entirely
       "out-of-sample", but multiple predictor scenarios are used in these cases, thus not being purely self-emulation. The "envelope"

option uses both SSP119 and SSP585, to construct a pattern using information from the extremes of the available dataset. The
       "furthest" option uses the most different scenario – SSP585 to emulate SSP119 and vice versa – an unlikely choice, but a test
       that can give information about the effect of a wide range of choices. "all others" utilises all of the available scenarios except
       the target one, and "nearest" uses the closest scenario in radiative forcing. The 2070-2100 multi-model mean errors associated
       with these eight cases are shown in Figure 9. To compare to inter-model variability, the magnitude of the errors minus the

inter-model standard deviation are shown in Supplementary Figure S11.
       The SSP119 error maps are remarkably insensitive to the choice of predictor datasets; the timeseries error intrinsic to SSP119,
       a substantial fraction of the error as shown in Figure 7, ensures a baseline error. Since the disequilibrium is qualitatively similar
       between early-late SSP119 (responsible for the timeseries error) and other scenarios to SSP119 (giving the pattern error), the
       distributions of the pattern and timeseries errors are similar. When targeting SSP585, however, a clear difference in the error

magnitude is found when using different predictor datasets. Low errors occur with the envelope method, with similarities to
       the self-emulation pattern suggesting SSP585 drives the pattern generated from the combination. The "furthest" case gives
       large errors as expected; "all others" and "nearest" give relatively similar, and smaller, errors.
       The patterns of comparison between the error magnitude and the inter-model deviation (Supplementary Figure S11) are very
       similar, with few areas of agreement between models. This is unsurprising for SSP119 given the similarity in the error, but for

SSP585 the substantially different error magnitudes and patterns result in similar, and small, areas of agreement in the pattern
       scaling error. This is presumably as the inter-model spread scales with the magnitude of the error, reflecting disagreement in
       the driving processes of the pattern scaling errors. This is again consistent with prior findings that inter-model variation is
       larger than pattern scaling errors (Goodwin et al., 2020; Herger et al., 2015; Osborn et al., 2016, 2018; Tebaldi & Arblaster,
       2014; Tebaldi & Knutti, 2018).






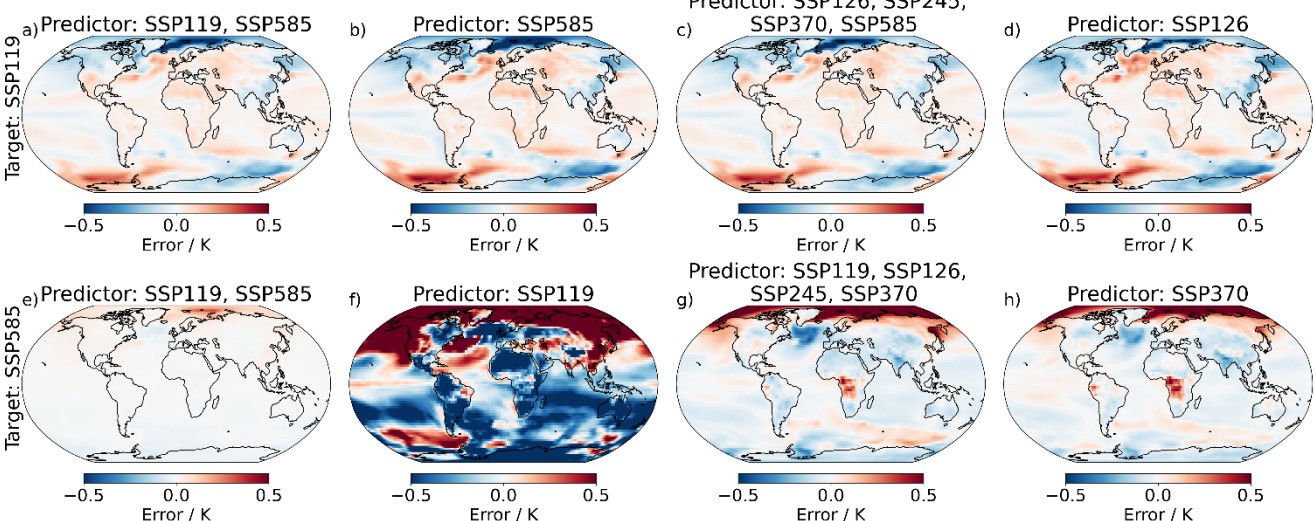

Figure 9: 2070-2100 pattern scaling errors when projecting SSP119 (top) and SSP585 (bottom) for patterns using four sets of predictors: envelope (left; SSP119 and SSP585), opposite (2nd column; SSP585 for targeting SSP119 and vice versa); all others (3rd column; i.e. the four scenarios other than that being targeted); and nearest (4th column; SSP126 to target SSP119 and SSP370 to target SSP585 - the nearest scenario to each target in RF terms).

## 5 Discussion and Conclusions

This study presents a decomposition of pattern scaling errors into two components: one due to differences in the pattern between the predictor and target datasets (the pattern error), and one due to internal nonlinearities in the target scenario (the timeseries error). The differences in warming patterns between pairs of single-forcer experiments and plausible future scenarios, causing the pattern error, were also investigated, along with case studies of the impact of the timeseries error, and the total impact on the application of pattern scaling to the SSPs was tested.

Self-emulation – using a scenario's pattern to reproduce its own response – generates zero spatial error by definition when averaged across the whole period. For a time period within the scenario, errors occur due to differences in the temporal shape of the local and global temperature. These differences are intrinsic to the scenario, and irreducible under simple pattern scaling. Here, spatial differences in the peak warming year in low emissions pathways was found to manifest in substantial emulation errors across regions and models.

When emulating out-of-sample, pattern errors are introduced due to the pattern differences between the scenarios, combining with the internal timeseries error in the target scenario. Robust differences were found between temperature change patterns under historical GHG and aerosol forcings, with the NHMLs more sensitive under aerosol forcing due to the historical predominance of aerosol emissions there. Differences between temperature change patterns under future scenarios were less significant between models, as found in prior work studying future emissions scenarios (Goodwin et al., 2020; Herger et al.,





2015; Osborn et al., 2016, 2018; Tebaldi & Arblaster, 2014; Tebaldi & Knutti, 2018). However, the difference resembled differences between transient and equilibrium patterns in prior work (Herger et al., 2015; Huang et al., 2020; King et al., 2020; T. D. Mitchell, 2003) – higher sensitivity over tropical land and lower over high-latitude oceans in SSP585 – indicating the
different warming rates in the scenarios are an important cause of difference between these scenario patterns. Aerosol concentrations are also substantially different between these scenarios, potentially causing a less-sensitive East Asian response under SSP585, though this wasn't robust between models.

The pattern error drives over 80% of the overall pattern scaling error when emulating a high-emissions scenario using a low-emissions pattern, causing pattern scaling errors to be strongly dependent on the predictor dataset used. In contrast, the
timeseries error contributes around half the error for emulating low-emissions scenarios, rendering the choice of pattern less important, though choosing scenarios closer in radiative forcing to the target still reduces the overall error.

Splitting the total error into these components allows for an understanding of the relative importance of the limitations of the assumptions which generate the errors. Understanding which source drives the error of a given pattern scaling application can guide efforts to reduce these uncertainties.
The errors associated with differing aerosol emissions and differing levels of warming, including stabilisation and relative cooling, will presumably be more important for the SSPs analysed here than the prior RCPs, which saw a narrower range in aerosol emissions and levels of warming. The tighter range in $CO_2$/non-$CO_2$ forcings ensured pattern scaling worked well under the RCPs (Goodwin et al., 2020), but variations in the aerosol pattern will lead to greater pattern scaling errors (Xu & Lin, 2017). Projecting existing and future Paris Agreement-consistent scenarios, i.e. those which stabilise temperatures below
2°C and reach net-zero GHG emissions by 2100 (Schleussner et al., 2022), such as the C1 and C2 scenarios in IPCC AR6 WGIII (Kikstra et al., 2022), will lead to issues related to equilibrium and transient pattern differences (King et al., 2020).

The efficacy of pattern scaling is constrained by the choices of patterns available, i.e. the dataset of scenarios simulated in multi-model ESM ensembles, which is itself determined by the trajectories chosen under projects such as ScenarioMIP. These might not cover the full relevant range of scenario attributes (Guivarch et al., 2022); there is a lack of stabilising and cooling
scenarios in the extant datasets (Tebaldi et al., 2022), with a recognised need for more equilibrium experiments in the future (King et al., 2021). However, this work demonstrates that scenarios with these properties are less amenable to emulation via pattern scaling than higher-emissions ones.

These results suggest that caution should be taken when applying simple pattern scaling to emulate low emission scenarios, as these are intrinsically less amenable to emulation via pattern scaling. Large differences in forcing pattern and rate of warming
between predictor and target scenarios also lead to substantial emulation errors.

This paper focused on annual mean temperature, but it would be useful to determine the relative roles of the pattern and timeseries errors for other variables, to determine the extent to which their emulation is limited by nonlinearities in the target scenario. The distribution of temperature variability is also crucial for impact analysis, can change under external forcing (Olonscheck & Notz, 2017; Pendergrass et al., 2017), and has been incorporated into emulation tools such as MESMER
(Beusch et al., 2020).

Only simple pattern scaling – using one predictor, global temperature, to emulate the local temperature response using a single pattern - was studied here. These errors may be mitigated to an extent by other methods, such as using patterns dependent on forcer (Xu & Lin, 2017; Kravitz et al., 2017; Schlesinger et al., 2000) or response timescale (Zappa et al., 2020). Improvements have also been found by adding extra predictors in addition to global temperature, such as land sea contrast (Herger et al.,
2015) and ocean heat uptake (Beusch et al., 2020).

Regional climate changes are key for understanding the impacts of different policy choices, but the links between global mean temperatures and their regional impacts haven't been fully explored within the IPCC framework (Kikstra et al., 2022). Emulation of regional impacts via tools such as pattern scaling, in a consistent framework such as MESMER (Beusch et al., 2020), provides a crucial means by which to estimate the local impacts of new emissions scenarios without the need to perform
expensive, time-consuming ESM simulations.

This paper furthers the understanding of pattern scaling by decomposing the errors into those attributable to different assumptions. Further research to reduce these errors where possible will be crucial to enhance our understanding of future climate change.

**Data availability**

The CMIP6 data utilised in this study are available for download at https://aims2.llnl.gov/search

**Author contributions**

All authors designed the study and reviewed the manuscript. CDW carried out the analysis and led the preparation of the
manuscript.

**Competing interests**

The authors declare that they have no conflict of interest.

**Acknowledgements**

The authors would like to thank Lea Beusch and Mathias Hauser for their useful discussions and input on MESMER and its implementation. The authors would also like to thank the modelling teams who designed and ran the models (see Supplementary Tables S1 and S2) and made the data available, and the World Climate Research Program for facilitating the CMIP6 process.


**Financial support**

CDW, ACM, PMF, and LSJ were supported by the European Union's Horizon 2020 programme under grant agreement No 820829 (CONSTRAIN). PMF was supported by the Vol-Clim NERC grant NE/S000887/1.



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
