# Peer review of "Understanding pattern scaling errors across a range of emissions pathways"

_EGUsphere, 2022_

## Author Comment (AC1)

**Reviewer #1**

Review of "Understanding pattern scaling errors across a range of emissions pathways"

Wells et al. consider errors in pattern scaling for different emission scenarios. They decompose the error into timeseries and pattern errors to better understand their sources. This is highly relevant due to the emergence of climate emulators used to estimate local impacts of emissions - often for scenarios the emulators were not originally trained on. Overall the manuscript is well written and clear. However, there are some points I would like the authors to clarify.

We thank the reviewer for their positive comments about our manuscript. We respond to their specific points below.

Main points

In the data part I missed that you calculate anomalies of the predictor and target variables and I also see no mention of the reference period. Please also explain how you deal with ensemble members. Do you use one or many per model? How do you estimate the local slope for models with many ensemble members? How do you avoid giving more weight to models with more ensemble members?

The reference period under each experiment is the first 50 years of that experiment; this has been added to the final paragraph of Section 2.1.

We use all members that were available on cmip6-ng; these are listed in Supplementary Tables S1 and S2, and we have made clearer reference to this now.

The pattern regression in MESMER is applied across all members of a given model-scenario combination simultaneously – again we have added this to the methodology section.

For inter-model results, we average over the individual model ensembles first, and compare these averaged results for each model; we've now noted this in the second paragraph of Section 2.1.

Please emphasize more that you use only part of the full MESMER emulator.

We have further emphasised this point by modifying the abstract and adding to Section 2.1

You mention several times that using patterns to extrapolate are worse than to interpolate and cite a number of studies showing this. However, I miss a citation of Beusch et al. (2022) who also discuss this. Further, the MESMER emulator has been already extensively evaluated (Beusch et al., 2020a, 2020b) and I think your paper would benefit from discussing this and showing how your paper goes beyond the state of the art.

We have added a citation to the 2022 paper showing that extrapolation generates poorer outcomes.

We think those useful papers address different questions to this manuscript. Beusch et al., 2020a studies self-emulation cases, with an emphasis on the internal variability component, and Beusch et al., 2020b applies the method to a process of combining output from different models to best reproduce observations. This paper focuses instead on applying the pattern scaling in the mean MESMER module to unseen scenarios, looking at how well the method might be applied to generate new scenarios, and the types of errors resulting from this application.

It's interesting to see that scenarios with a peak in the global mean temperatures show local time lags and would profit from additional predictors. Can you speculate how much the missing MESMER components (i.e. the auto regression) would help alleviate this problem?

Since the AR(1) process in the full MESMER system is applied to the local variability of an emulation, and we are interested in the mean error over period of decades, we think there should be no difference from including this module. The same applies to the other modules which focus on internal variability; the relevant component for long-term mean responses is the local trends component which we use.

Consider changing the way you show significance. I was first confused why you would subtract the standard deviation from your difference signal - I overread the word "magnitude". Therefore I suggest you do one of the following:

(i) switch to showing significance with a test statistic and hatch the non-significant areas in your plots. This should reduce the number of figures and plots without losing (much) information. (E.g. by using a Wilcoxon Mann-Whitney U test and accounting for the large number of conducted tests, by applying the approach of Benjamini and Hochberg (1990), see also Wilks (2016)).

(ii) If you keep your current approach I strongly suggest to make it more clear - add vertical bars in the title of the figures to make it clear that it is the magnitude of the difference and also explain what values larger, smaller and almost equal to zero mean at around L210.

(iii) Instead of subtracting the standard deviation could you divide by the inter model standard deviation. That would seem more intuitive to me.

We agree that the significance could have been more clearly shown. We have followed your suggestion (iii), by dividing by the intermodel standard deviation. We have then applied stippling to show where the magnitude of the difference is greater than 1. We have applied these changes to the similar figures in the supplement.

Minor Points

L131: Why is "pattern scaling more accurate than the timeshift method"? Wouldn't the latter allow for non-linearities?

The Tebaldi et al., 2020 study cited in this section finds that pattern scaling performs better for emulating the mean response for out-of-sample scenarios; we have updated the text to clarify these specifics. Nonlinearities will be recorded using the timeshift method for analysing given warming levels within a scenario, but when trying to generalise to find the response at a given warming level across scenarios, different nonlinearities between scenarios (from differences in e.g. warming rate and forcing patterns) will cause errors.

L143: The intercept will also depend on how the anomalies are calculated (and how ensemble members are treated).

This is a good point, and we have now noted this; the intercept should in theory still be zero.

L212: Explain that the pattern averages to 1 globally per design and only because of the investigated variable is tas.

This is an important clarification and we have now added this.

L245: "pattern difference is not as robust between models" that is an interesting way to put it. Isn't it good for pattern scaling if there are few regions with strong differences?

It is correct that similarities between future scenario patterns demonstrate the efficacy of pattern scaling, but it also reflects intermodel disagreement; we have added to this sentence to highlight these points.

Figures

General: many of the color scales you show saturate on a large part of the maps. Consider widening the shown range to allow distinguishing the patterns better. Please write the labels and units as "Error (K)" instead of "Error / K". Then it looks less like a division.

We have made several scales larger, and removed the misleading "/" in labels in all relevant figures in the main text and supplement.

Figure 1: I appreciate that you showcase the different errors in an example. However, I think using a scenario that is symmetric in its global temperature makes it more difficult to understand than necessary. Consider showing a non-symmetric scenario, e.g. just increasing the temperatures from 1°C to 2°C until the end of the century.

We agree that it would be more helpful to use a purely increasing temperature scenario, but we found that using asymmetric trajectories led to local regression parameters that did not quite match the global response when applying a timeseries anomaly. We have tried some different ways of approaching this but were unable to improve on the original schematic's scenarios.

You could also consider switching the first and second columns. If I understand this correctly the (current) middle column is the "forcing" for the emulator while the (current) first column is the "response", so switching them could help clarify this relationship.

The orange line on the middle panels is the actual trajectory, so it may be useful to be able to show this first. But the dashed blue line on the middle panel is the emulation itself, which is produced using the regression shown on the left panel, and therefore ideally will come afterwards. So we feel on balance that the original format best shows: the method (i.e. calculating regression parameters) -> output (the emulation as compared to the actual trajectory) -> error. One option would be to add an initial column showing only the original trajectory but this would repeat the orange line in the current middle panel and increase the number of panels substantially. We have reworked the discussion of this figure to better communicate these stages.

Figure 2 and 3: Panels a) and b) don't have a diverging scale and should therefore not feature a diverging colormap. Please use one with a sequential color map. (If you want to emphasize deviations from 1 you can keep a diverging color map but you should mention this and also use another color map as for c) and d)) Depending on what you decide on showing significance, also consider changing the colormap of d) to indicate it shows something else than c).

We have changed the text as suggested to mention the significance of the sign of the divergence from 1 in the pattern, in the top row. We have increased the scale to avoid saturation in each panel. We have changed the colourmap in panels c and d as also suggested. For the fourth panel we have

divided by the standard deviation as suggested instead of subtracting, and also changed the colormap as suggested. We have added stippling to the fourth panel to note differences larger than 1 intermodel standard deviation. We have applied these changes to the related supplementary figures.

Figure 4. I'd be interested to see how similar the pattern in b) and c) are, the saturation in b) makes this difficult.

The patterns in b) and c) should be similar, as both these errors scale with the difference between the predictor and target scenario patterns, which here are hist-aer and hist-ghg, with the predictor and target roles switched between the panels. We endeavoured to keep the scale consistent between all four panels, to emphasise the difference in error size between self-emulation (the diagonals) and out-of-sample emulation. We experimented with log scales, but this didn't aid the interpretation. We instead show the difference between 4b and c, and their ratio:

[Figure]

The errors are the same sign generally (right), with the difference matching the pattern differences as expected (left). We have included this figure in the supplement (now Figure S4) to aid readers in gauging the difference between these patterns.

Figure 7: I suggest you label the "Target" below the axes and to maybe not rotate them by 45° (they might just have enough room) - up to you. Please add % as units to d).

We have added the Target labels to the bottom of the axis, but removing the rotation left too small a gap between the labels. We have added the missing % to the d label.

Figure 8: The black vertical lines described on L416 are missing.

Many thanks for spotting this – we have added the lines.

Text

Thanks for these useful alterations; we have applied them with the exception of one which we have expanded on below.

L7: delete "multiple"

L7: delete "a few"

L26: Expand "IPCC AR6 WG1"?

L38: Maybe delete "change"

We feel that simply removing this word would give a different meaning, as we are talking about impacts of climate changes here; we have reworded instead to clarify this.

L80: Expand "RCP" and explain what this is.

L85: "than the RCPs" -> "than any of the RCPs"?

L84: "remains to be done" consider rewriting

L100-L104: Make it clearer that these are your two assumptions (e.g. turn it into a list or add (i) and (ii)).

L102: timeseries -> temporal

L103: simply modified -> scaled

Section 2.1: I highly recommend to split this into two sections one on the data and one on MESMER.

L136: against the smoothed -> against smoothed

L137: parameter -> "slope" or "scaling factor"

L137-L138: This … SSP119: The sentence sounds off.

L205: You explain the pattern in panel b) first. Consider reordering.

We feel that it is best to discuss the GHG pattern first, as it is simpler to understand the climate response to uniform forcing, with the aerosol pattern as a modified version of this. But then with panel c we feel it makes most sense to show aerosols minus GHG, as this is more readily understood with reference to the aerosol effect. Changing the panel order of a and b would make the sign of c less intuitive, so we feel on balance that it should stay in the current order.

L349: Upon even only -> Even for

L419: Consider rewriting the introductory sentence.

References

Hochberg, Y. and Benjamini, Y. (1990), More powerful procedures for multiple significance testing. Statist. Med., 9: 811-818. https://doi.org/10.1002/sim.4780090710

Beusch, L., Gudmundsson, L., and Seneviratne, S. I.: Emulating Earth system model temperatures with MESMER: from global mean temperature trajectories to grid-point-level realizations on land, Earth Syst. Dynam., 11, 139–159, https://doi.org/10.5194/esd-11-139-2020, 2020a.

Beusch, L., Gudmundsson, L., & Seneviratne, S. I. (2020b). Crossbreeding CMIP6 Earth System Models with an emulator for regionally optimized land temperature projections. Geophysical Research Letters, 47, e2019GL086812. https://doi.org/10.1029/2019GL086812

Beusch, L., Nicholls, Z., Gudmundsson, L., Hauser, M., Meinshausen, M., and Seneviratne, S. I.: From emission scenarios to spatially resolved projections with a chain of computationally efficient emulators: coupling of MAGICC (v7.5.1) and MESMER (v0.8.3), Geosci. Model Dev., 15, 2085–2103, https://doi.org/10.5194/gmd-15-2085-2022, 2022.

Wilks, D. S. (2016). "The Stippling Shows Statistically Significant Grid Points": How Research Results are Routinely Overstated and Overinterpreted, and What to Do about It, Bulletin of the American Meteorological Society, 97(12), 2263-2273.

**Reviewer #2**

General Comments:

The paper by Wells et al. presents an analysis of the errors arising when using the mean component of MESMER for emulating climate model simulations for different emission scenarios. The paper is well structured and the analysis appears sounds, and therefore I find it eligible for publications after revision.

We thank the reviewer for their detailed comments on the manuscript which we respond to below.

I think it could generally be rewritten more concisely as there are detailed descriptions of results that are sometimes trivial, and description of figures that would be better placed in figure captions. In addition, I think the motivation and usage of the model could be better explained. Particularly, I understood from Beusch et al. (2020) that MESMER served to emulate a single model-scenario (i.e. self-emulation) to general a large ensemble, e.g. we have model X with scenario SSPyyy and we emulate this single run to obtain a large ensemble with random internal variability. What is thus the purpose of understanding the cross scenario errors presented here? Is the goal to use MESMER for making regional projections? In which case, what is the point of using the 2020-2070 period to set up the emulator and then project the 2070-2100 period? Wouldn't we want to rather use historical and/or idealized simulations or observations to set up our emulator and analysed the errors induced by the historical pattern on emission scenarios (e.g. Geoffroy & St-Martin, 2014; Hébert & Lovejoy, 2018)? I initially expected that the aerosol and GHG patterns would be used to calibrate a two-pattern emulator, but rather we only get insights on the difference between the extrapolation of these two patterns, results which I thought were a bit trivial since we already know that aerosols have a localized impacts. Wouldn't it be possible to use those two patterns to emulate future scenarios if we have a decomposition of global mean temperature into aerosol and GHG driven components? I think this would be a more powerful framework since we could then use those patterns to emulate any scenarios given the global mean temperature along with the aerosol and GHG forcing timeseries. It is not necessary for the authors to do this in this paper, but I wanted to outline what I think would be useful to broaden the scope of the study.

Beusch et al. 2020 do focus on comparisons of self-emulated ensembles based on internal variability, but the full motivation of this kind of research is to ultimately generate out-of-sample approaches, as explored in the Beusch et al. 2021 paper coupling to the MAGICC global mean emulator. The application of emulators to generate out-of-sample scenarios is motivated by, for example, WGIII of the IPCC. A closer examination of the types of fundamental errors that can occur in applying this method to new scenarios (which can be explored by looking between existing scenarios) is what we seek to address with this manuscript. We could have used a historical baseline to undertake this, but we felt it best to reduce the amount of scenarios brought to the reader, and since we were interested in the types of difference between future scenario patterns and their relative effect on future emulation. The aerosol/GHG decomposition is very interesting, and has been applied by some

studies that we already cite (Kravitz et al., 2017; Xu & Lin, 2017; Schlesinger et al., 2000), but still suffers from issues of non-constant aerosol forcing pattern, as well as other effects related to thermal inertia. We have added to the introduction (L30) to highlight this motivation for out-of-sample exploration of novel emissions scenarios. We have also moved figure descriptions to their captions where possible to make the text more concise, and made edits throughout to further improve the concision.

Specific Comments:

Line 47: "forcer pattern" --- I'm unsure about the use of 'forcer' here and elsewhere, shouldn't it be 'forced pattern'?

We have changed this wording to "forcing pattern", since it is the difference in the distribution of the actual forcing which causes the differences discussed here. This is consistent with the wording used at other points in the manuscript. In other occasions where "forcer" is used, this is to discuss the actual forcer set i.e. GHGs vs aerosols.

Line 129: "This study utilises the mean response component of the MESMER model (Beusch et al., 2020), implementing pattern scaling to emulate the spatial annual mean temperature response in a scenario." --- Is it still the MESMER model if we use only the mean component? Then isn't it just a regression of the local temperature with respect to the global one?

The mean response component implements the simple pattern scaling here; this is the component we are applying, as we are studying the long-term response. The internal variability components would not deviate from this when averaged over sufficiently long periods. So it is just local/global regression. This paper isn't aimed at evaluating MESMER specifically, but at studying the implications of the pattern scaling assumptions more generally. We have now further clarified, including in the abstract, that we use only the mean response part of MESMER.

Line 137: "This is performed to ensure the global average parameter is very close to 1 K/K, as it should be by definition, when predictor the model on an individual low-emission scenario such as SSP119." --- What global average parameter are we talking about? The global average of the local sensitivities? Why does it matter to smooth or not the local temperature for the regression to obtain an average close to 1? Also, review the formulation of sentence, 'when predictor the model' doesn't sound right. I would also explain here or somewhere else the units of K/K since at first one thinks why don't they just cancel, and well they do, but I understand you wanted to make explicit that this was a local sensitivity of the local temperature to the global one, right?

Yes, in this we are referring to the global average of the local sensitivities, which does have units of K/K which cancel. This paragraph was poorly worded and we have rephrased it to clarify this. Regarding local vs global smoothing, for a low-emission scenario in which temperature peaks and falls, smoothing the global response while not smoothing the local response acts to slightly enhance the peak values locally, such that when the regression is applied to these values, the global parameter can be pulled above 1. Just to visualise this, in the below left figure a schematic scenario is shown in green which warms and cools linearly. The red dashed line shows the smoothing of this.

[Figure]

If we take the scenario to be spatially constant, i.e. each location exhibits the same response (and hence this is the global mean), then regressing the raw local data on the smoothed global data leads to the following regression:

[Figure]

The black line shows the fit which arises from this – rather than being linear with a gradient of 1, as in the scenario, it has risen slightly. We found this made a difference to the low-RF scenarios as they exhibit this peak warming. We have now added some sentences to the manuscript to clarify this, though we felt it unnecessary to go into as much detail as in this reply.

Line 145: "A given emulation consists of the predictor set – comprising one or many scenarios – and a target scenario." --- I think this could be better explained. Are we talking about a set of model simulations following certain emission scenarios that are used to estimate the pattern, and then one separate scenario with its own set of model simulation is used as target?

We have split this into two sentences now; your understanding of our meaning is correct and we have reworded the section to better present this point.

Line 156: "temperatures relative to pre-industrial times rise from 1 K in 2015 to 2 K" --- Maybe give the approximate year when the temperature reaches the 2k 'from 1k in 2015 to 2k in ????'

We have added this information.

Line 208: "In hist-aer, the land-ocean distinction is still clear, but the northern hemisphere land is particularly sensitive, due to the historical concentration of aerosol emissions within this region." --- Sensitive isn't exactly the right word right? Are the land region really more sensitive, or is it purely because of the higher aerosol emissions there that the regression slopes are higher? I would consider rephrasing the paragraph to clarify this.

We agree that this wording was not ideal, and have replaced uses of "sensitive" with alternative phrasing, to reflect that these are differences in the regression parameters due to the different pattern of forcing.

Line 217: "Parts of the NHMLs exhibit a significantly more sensitive response to hist-aer, including the USA, Europe, and east Asia, and the Southern Hemisphere oceans are significantly less sensitive." --- Again, this sounds like sensitivity to aerosols is a local property of the system, but really, the pattern of the response just corresponds to the sources of aerosol emissions. This would likely be outside the scope of this study as it might require more data about the spatial distribution and dispersion of aerosols, but it would be interesting to quantify the actual sensitivity to aerosols taking into account the pattern of emissions (and their dispersion).

Agreed that this is convoluted, and that this would be interesting to dig into but would require a concerted effort. We have rephrased this section to clarify this point.

Figure 2ab,3ab: I'm not sure the divergent colour palette is appropriate since there is no fundamental difference between values below 1 and above right?

The difference around 1 signifies the regions which see a higher/lower temperature response than the global mean. This allows a clearer comprehension of the qualitatively different response over different regions – e.g. the land/sea contrast in hist-GHG. The other reviewer noted this and suggested they could be left as divergent if this reason was noted; we have added this to the text.

Figure 2d,3d: Wouldn't it be more informative to look at the ratio of the absolute difference over the inter-model spread?

The other reviewer also noted this, and we have changed to divide by the inter-model deviation instead of subtracting. We have used the actual difference rather than the absolute, so the reader can see clearly the sign of the deviation. We have applied these changes to the relevant supplementary figures too.

Line 228: "Figure 3 shows the same analysis for SSP119 and SSP585 in a similar way to Figure 2." --- I would complete this sentence with a restatement of what is calculated, something like, if I understand well: '...similar way to Figure 2, i.e. the local temperature series are regressed with the global mean temperature to extract a local sensitivity to global temperature changes.' (could be written more concisely).

We have changed and extended this sentence to explain this further.

Line 251: "Clear, significant differences are therefore found between the temperature response patterns attributable to different historical forcers, consistent with their different spatial patterns." --- I hate to be this guy, but if you say significant, the reader expects a p-value, and you should explain the statistical test used, the null hypothesis considered, etc.

This is a fair point, and we have removed the word "significant", drawing attention instead to the greater difference than the inter-model standard deviation across broad areas. We have also rephrased other occurrences of this wording.

Figure 3: What period is used to train the pattern? In the methods it is said that the first 50 years are used, but I don't think it is said which period the SSP simulations cover. In any case, I think it would be helpful to explicitly state the tiem period sued for training the model.

The patterns in Figure 3 are regressed across the full period, with anomalies relative to the first 50 years. We have added information on the time domain of each experiment in L125, and information on the regression data in L137.

Line 271: "since the aerosol pattern is more sensitive here than the GHG response, and the Southern Ocean is conversely under-sensitive" --- Again, I'm really not convinced by the usage of sensitivity when it comes to the aerosol pattern. It's not about the sensitivity of aerosols, but rather the strength and spatial distribution of the aerosol forcing. The Southern Ocean is not less sensitive to aerosol forcing, there are just much less aerosol emissions reaching that region.

We have rephrased this to remove this ambiguity; we have also reduced the use of this wording throughout other sections of the manuscript, to avoid relying on this wording. The occurrences we have left in feature explanations of the effect (referring to the aerosol emissions pattern) or other physical responses (such as saturation of feedbacks).

Line 305: "Errors are significant in the out-of-sample emulations" --- Again, if you say significant, we expect a p-value, if you don't want to give a p-value, use larger instead, otherwise we would also like to know if the smaller errors of the self-emulation are significant or not, just because they are smaller doesn't mean they might not be significant.

As per an earlier comment, we have removed uses of "significant" to remove this ambiguity.

Line 358: "Note the smaller scale on the timeseries error plot." --- Might be more useful to have this statement in the caption.

We have made this change.

Line 365: "note the slight variations in the SSP119 column compared to the SSP585 one." --- Do you mean 'smaller' variations rather than 'slight'?

We have used this improved wording.

Line 387: "The patterns are similar between SSP119 and SSP126, indicating some consistency between scenarios in this effect." --- Why are only those two scenarios considered for this comparison? Wouldn't it also be interesting to see the pattern for SSP245 with a later peak and drop?

The peak warming pattern for SSP245 is 2100 everywhere (i.e. continuous warming) except in the North Atlantic for a couple of models. We applied a 30-year smoothing to the data before taking the peak; using a smaller window might give earlier peak warming, but would also introduce more noise to the data. We have added a sentence to this section to note this fact.

Figure 5: Unclear on what period the patterns used for emulation were calculated.

We have now clarified the periods used for the pattern calculations as noted above.

Line 390: A lot of this paragraph could belong to the caption instead. There were several such instances where the figure was described in the text rather than in the captions, I would consider improving the captions and shortening the text to the results only and avoiding the description of the figures there.

We have edited this paragraph, moving information to the caption, and done the same for other figures in the manuscript.

Line 390: I would motivate why those specific region-model-scenario are used, I guess simply to explore problematic behaviours?

Yes, these combinations showcase the different types of error which occur; we have clarified this sentence to note they were chosen for this purpose.

Figure 9: What period is used to train the predictor?

The patterns regressions are applied on data relative to the first 50 years of the scenario; this is used to baseline the ESM output and emulations too. We have clarified this point in the manuscript.

---

## Author Response (AR2)

We thank the editor for the additional comments on the manuscript; our response to each comment, and details of further changes made, are below.

Overall the paper is much improved and the authors have addressed reviewer comments.

However, there is a significant clarification that needs to be made. The authors have clarified in separate correspondence that: "the method is always applied on each model separately. So the pattern for a given scenario (or set of scenarios) is found for each model separately, and then this model-specific pattern is multiplied by the GMST for the target scenario from that same model, to get a model-specific emulation. Then in the plots the multi model means are taken - and for uncertainty analysis this is compared to the inter-model deviation."

Please make sure the methods section clarifies this, this should be stated in more than one place so that this point is not missed. In particular provide more detail on the methodology in terms of what steps were taken in what order, since this will presumably make a difference in the results.

This does raise some questions. The paper, therefore, seems to be focusing on decomposing pattern scaling of the multi model mean pattern. This, therefore, seems to result in the paper focusing on areas where models agree in the pattern of change. Where models disagree, the mean change will be smaller and less signifiant. In this sense, the current conclusions in the paper seem to be somewhat limited - they don't apply to GCM patterns in general, but pertain more to those areas of the globe where there is robust agreement between GCMs.

This is contrast to evaluating how the pattern scaling method works for individual climate models, and then presenting analysis of how well the method works across models.This is likely a more common application of pattern scaling. For example, most impact analyses use not mean patterns of change across models, but patterns for specific models (usually repeating the analysis for multiple models). Can the results be analyzed to also determine if the conclusions of this work are similar for individual GCM patterns in terms of the decomposition of results into the two components? Are the conclusions robust across different GCMs? Can the analysis and results be used to address this point?

The lack of clarity in our draft manuscript has led to some confusion on this point.

We apply all the analysis on individual models, including calculating pattern scaling errors. We only take multi model means when we are plotting, and then use the intermodel standard deviation of the results being plotted to give a sense of the level of agreement amongst models.

For example Figure 4 shows the multi model mean pattern scaling errors - i.e. first we do the pattern scaling for each model, then we calculate each model's error by subtracting the ESM data from each emulation separately, and then we average the error across models. And in Figure 6, the time-series of the "Timeseries" and "Pattern" errors are calculated for each model separately, and then the multimodel mean is plotted in the solid line, with the shading showing 1 standard deviation across the models, showing the results are robust across models.

We decompose the errors separately for each model before comparing. We do analyse where geographically the models agree/disagree on the pattern itself, which we feel is an important question to ask, but we don't average across models until the very end.

To address this lack of clarity we have edited the methods section to better present the methodology. In particular we have added a paragraph at the end of section 2.2 to clarify the application to individual models. We have also added to the conclusions to emphasise this point.

Specific comments

Section "2.2 Pattern Scaling Methodology". Current version is largely one large paragraph, which is difficult to read. Break this up (and add the material mentioned above).

We have split this large paragraph into several smaller ones to aid reading, as part of the editing process above.

Line 141

"Inter-model results are averaged first over each model ensemble, with this model average then compared, to avoid weighting by the ensemble size of each model."

This implies that for models with more ensembles, patterns are likely more robust, with inter-ensemble noise averaging out across ensemble members. While for models with fewer ensembles, patterns are less robust (larger regression uncertainty, all else being equal.) How does this impact the results?

This is an interesting question; we did some analysis of the sensitivity of the pattern to the ensemble size, and found that >=3 members was generally enough to give a robust pattern over most areas. We might look to include this analysis in a future paper and explore this further, but we don't think it justifies inclusion in this paper. Most of our ensembles were larger than 3 so this variability likely wouldn't drive a huge variation in our results, but some ensembles had only one member so there will be an effect. We have added a sentence to the discussion noting the range in ensemble sizes and suggesting this might be useful to investigate further in additional work.

This leads to another question, it appears that regression fitting uncertainty is not used in this analysis? Or is it?

We don't analyse the regression uncertainty in the presented results here; since pattern scaling assumes linearity, we test the effect of this nonlinearity error by comparing to the ESM data, rather than analysing the error in the regression itself.

Line 160: "For calculation of the pattern in a model with multiple ensemble members, MESMER applies the regression across the data from all the members simultaneously."

clarify what this means. Is the regression performed for each ensemble member and then somehow averaged? (Are the coefficients of the regression equation averages of the coefficients for each ensemble member? or something else? How are non-zero intercepts treated.)

We have clarified this section – the regression is applied once, on the concatenated members. Any intercept is added to the emulation, noted on lines 160 and 197.

Line 199

Here

"The timeseries error, in row two of Figure 1, ..."

Should this be column 2? (e.g. isn't every panel in column 2 the time series error?).

Not exactly; what we term the "Timeseries error" is shown in the 2nd row in Figure 1; row 3 shows the "Pattern error".

Please list in the SI the exact scenario data members used (either by DOI or other identifier), along with the data of access of the cmip6-ng database. This is necessary to assure your results can be, in principle, replicated. While the CMIP6 model data is relatively stable at this point, changes and new submissions may occur, so it is necessary to document the exact data used.

We have replaced Tables S1 and S2 with a single Table S1 which lists all ESMs, the scenarios for which they are used, and the individual members used for these, using the ripf codes.

Line 220 this "Figures 2a and b shows the multi-model mean hist-aer and hist-GHG response pattern based on regression across the whole period (1850-2020)". is unclear.

Define how the multi-model mean regression is derived in the method section.

We hope this has been clarified by our additions to the methodology. The pattern is calculated for each model separately (i.e. the regression is applied to 1850-2020 for each model separately), and the multi model mean of these patterns is shown.

line 495 -typo

-> that of the scenario being emulated

Many thanks for spotting this – we have corrected this typo.

---

## Author Response (AR3)

We thank the editor for the additional comment on the manuscript; our response describing the additional change made is below.

Expand description of regression method to clarify treatment of multiple ensemble members.

We have clarified the methodology in Section 2.2 by adding sentences describing the approach for multiple members.